Eurasian jays do not copy the choices of conspecifics, but they do show evidence of stimulus enhancement

Miller Rachael 1
Logan Corina J. cl417@cam.ac.uk 2
Lister Katherine 2
Clayton Nicola S. 1
1 Department of Psychology, University of Cambridge , Cambridge , United Kingdom
2 Department of Zoology, University of Cambridge , Cambridge , United Kingdom
Vonk Jennifer
Electronic publication date: 2016 Dec 1
Publication date: 2016
Volume: 4
Electronic Location ID: e2746
Received 2016 Sep 30; Accepted 2016 Nov 2
Copyright: ©2016 Miller et al.
Copyright year: 2016
Copyright holder: Miller et al.
License: This is an open access article distributed under the terms of the Creative Commons Attribution License, which permits unrestricted use, distribution, reproduction and adaptation in any medium and for any purpose provided that it is properly attributed. For attribution, the original author(s), title, publication source (PeerJ) and either DOI or URL of the article must be cited.
License URL: https://creativecommons.org/licenses/by/4.0/

Keywords: Eurasian jay, Corvid, Social learning, Colour discrimination, Object-dropping task

Funding: European Research Council under the European Union’s Seventh Framework Programme FP7/2007-2013 ERC 3399933 Leverhulme Trust and Isaac Newton Trust with a Leverhulme Early Career Fellowship RM and NSC received funding from the European Research Council under the European Union’s Seventh Framework Programme (FP7/2007-2013) / ERC Grant Agreement No. 3399933, awarded to NSC. CJL was funded by the Leverhulme Trust and Isaac Newton Trust with a Leverhulme Early Career Fellowship. The funders had no role in study design, data collection and analysis, decision to publish, or preparation of the manuscript.

==============================
Corvids (birds in the crow family) are hypothesised to have a general cognitive tool-kit because they show a wide range of transferrable skills across social, physical and temporal tasks, despite differences in socioecology. However, it is unknown whether relatively asocial corvids differ from social corvids in their use of social information in the context of copying the choices of others, because only one such test has been conducted in a relatively asocial corvid. We investigated whether relatively asocial Eurasian jays (Garrulus glandarius) use social information (i.e., information made available by others). Previous studies have indicated that jays attend to social context in their caching and mate provisioning behaviour; however, it is unknown whether jays copy the choices of others. We tested the jays in two different tasks varying in difficulty, where social corvid species have demonstrated social information use in both tasks. Firstly, an object-dropping task was conducted requiring objects to be dropped down a tube to release a food reward from a collapsible platform, which corvids can learn through explicit training. Only one rook and one New Caledonian crow have learned the task using social information from a demonstrator. Secondly, we tested the birds on a simple colour discrimination task, which should be easy to solve, because it has been shown that corvids can make colour discriminations. Using the same colour discrimination task in a previous study, all common ravens and carrion crows copied the demonstrator. After observing a conspecific demonstrator, none of the jays solved the object-dropping task, though all jays were subsequently able to learn to solve the task in a non-social situation through explicit training, and jays chose the demonstrated colour at chance levels. Our results suggest that social and relatively asocial corvids differ in social information use, indicating that relatively asocial species may have secondarily lost this ability due to lack of selection pressure from an asocial environment.

Introduction

A wide range of corvid species (e.g., crows, ravens, jays) are known for their complex cognitive abilities, which are hypothesised to have been present in their common ancestor, thus forming a ‘general cognitive tool-kit’ across this taxa (Emery & Clayton, 2004). For example, even though rooks (Corvus frugilegus) do not make or use tools in the wild, they are able to spontaneously innovate these behaviours in the lab (Bird & Emery, 2009b). Further, there is evidence that some corvid species show cognitive competence across a wide range of social, physical and temporal tasks. For instance, California scrub-jays (Aphelocoma californica) show proficiency in cognitive tasks relating to memory (Clayton & Dickinson, 1998), future planning (Clayton, Emery & Dickinson, 2006; Raby et al., 2007), and social cognition through cache protection tactics (Clayton, Dally & Emery, 2007). As another example, rooks, in addition to their tool abilities, cooperate with each other to solve novel problems (Seed, Clayton & Emery, 2008) and appear to understand support relationships because they look longer at impossible scenarios (e.g., a ball suspended in mid-air rather than sitting on a table; Bird, Emery & J., 2010). Additionally, New Caledonian crows (Corvus moneduloides) reason about hidden causal agents (Taylor, Miller & Gray, 2012), reason by exclusion (Jelbert, Taylor & Gray, 2015), and learn socially about novel foraging problems (Logan et al., 2016a).

It is unknown whether this cognitive tool-kit includes the ability to use social information specifically in the form of copying the choices of others, which is distinct from changing behaviour when solving problems in different social contexts (several examples are given below). The corvid common ancestor is hypothesised to have been social (Clayton & Emery, 2007). If this assumption is correct, rather than the common ancestor being asocial with sociality having evolved several times in extant lineages, then there is reason to expect that relatively asocial corvids could have retained the capacity to use social information. For example, it could be adaptive by improving foraging and mate searching efficiency (e.g., Valone & Templeton, 2002). Alternatively, this ability could have been secondarily lost because of the lack of selection pressure from an asocial environment, in a similar manner to the secondary loss of caching (food hiding) in jackdaws (Corvus monedula; De Kort & Clayton, 2006). For example, in the absence of conspecifics for most of the year, there might have been an increased selection pressure to rely solely on personal information when foraging.

Most studies of corvid social information use, in the form of copying the choices of others, have occurred in social species (species that live in groups of at least pairs year-round), which makes it difficult to determine whether this ability is part of their general cognitive tool-kit. Evidence of social information use, specifically copying the choices of others, has been found in social corvid species, including pinyon jays (Gymnorhinus cyanocephalus; Templeton, Kamil & Balda, 1999), rooks (Dally, Clayton & Emery, 2008), jackdaws (Corvus monedula; (Schwab, Bugnyar & Kotrschal, 2008a), common ravens (Corvus corax; Fritz & Kotrschal, 1999; Schwab et al., 2008b), carrion crows (Corvus corone corone, C. c. cornix; Miller, Schwab & Bugnyar, in press) and New Caledonian crows (Logan et al., 2016a). Social species are predicted to be better at acquiring new skills in a social context than in a non-social context (Lefebvre & Giraldeau, 1996), because they may attend more to conspecifics than asocial species (Balda, Kamil & Bednekoff, 1996).

However, we are aware of only two tests of social information use in the form of copying the choices of others in a relatively asocial corvid. Clark’s nutcrackers (Nucifraga columbiana) did not learn a motor or a discrimination task faster in a social learning condition than in an individual learning condition, indicating that they did not use social information (Templeton, Kamil & Balda, 1999). This was in contrast with highly social pinyon jays that did learn faster in the social learning conditions (Templeton, Kamil & Balda, 1999). Additionally, Clark’s nutcrackers more accurately recovered caches they made rather than caches they observed others make, in contrast with social Mexican jays that were accurate in both conditions (Bednekoff & Balda, 1996). These results suggest that relatively asocial corvids attend less to social information than social corvids.

Outside of corvids, social learning in the form of copying conspecifics has been found in a number of asocial species including red-footed tortoises (Geochelone carbonaria; Wilkinson et al., 2010), black river stingrays (Potamotrygon falkneri; Thonhauser et al., 2013; Garrone Neto & Uieda, 2012), bearded dragons (Pogona vitticeps; Kis, Huber & Wilkinson, 2014), and in juvenile, but not adult, golden hamsters (Mesocricetus auratus; Lupfer, Frieman & Coonfield, 2003) and eastern water skinks (Eulamprus quoyii; Noble, Byrne & Whiting, 2014). These non-corvid species are likely to have had asocial ancestors, which suggests that social cues are not costly to attend to and can evolve outside of a social context in these taxa. However, at present, the sample size of the relatively asocial corvid species is too small to draw general conclusions about the influence of a corvid’s social system on their use of social information.

We addressed this gap by investigating whether the relatively asocial Eurasian jays (Garrulus glandarius) used social information provided by a conspecific. Eurasian jays do not live in social groups except during the breeding season when mated pairs defend a territory (Goodwin, 1951; Snow & Perrins, 1997; Clayton & Emery, 2007). There is evidence that socially housed Eurasian jays attend to social context to modify their caching and mate provisioning (courtship feeding) behaviour. For example, they prefer to cache in quiet rather than noisy substrates when in the presence of conspecifics that could hear but not see the subject (Shaw & Clayton, 2013); they attend to spatial and auditory cues when competitors are caching to later pilfer those caches (Shaw & Clayton, 2014); and subordinates inhibit caching in front of dominants and prefer to cache in less exposed areas (Shaw & Clayton, 2012). They also adjust their behaviour appropriately depending on whether they are caching or pilfering (Shaw & Clayton, 2014), and whether they compete with a dominant or subordinate (Shaw & Clayton, 2012). Furthermore, they prefer to cache out-of-sight behind an opaque barrier and at a distance when observed by conspecifics (Legg & Clayton, 2014; Legg, Ostojić & Clayton, 2016). During the breeding season, males are attentive to which foods their mates might prefer based on how much of which foods she has already eaten (Ostojić et al., 2013; Ostojić et al., 2014).

These jays were socially raised and housed, which differs from their relatively asocial system in the wild. The artificially social environment likely enhances their utilisation of any innate social skills because these skills will have been given the opportunity to develop from an early age. Therefore, if social skills are found in these conditions, it demonstrates the potential flexibility of this species to use social cues (if social cues are used). As such, the social capacities shown by socially raised and housed jays might differ from wild individuals. Despite the evidence that socially housed Eurasian jays can respond to social context in caching and mate provisioning paradigms, no study has yet tested whether this species uses social information to copy the choices of others, which could be useful for learning about foraging opportunities even in a relatively asocial species.

We tested whether socially housed Eurasian jays would use social information from a conspecific demonstrator when learning to solve a novel problem—an object-dropping task where an object must be dropped into a tube to release a food reward from a collapsible platform. Further, if the birds did not use social information to solve the task, we tested whether there was any evidence that they had attended to the demonstrator (as indicated by differences between groups with differing levels of social learning opportunities), and what they might have learned during this exposure. The object-dropping task has been used previously during pre-test training for Aesop’s Fable tasks in this species (Cheke, Bird & Clayton, 2011) as well as in a number of other bird species (rooks: Bird & Emery, 2009a; New Caledonian crows: Jelbert et al., 2014; Logan et al., 2014; California scrub-jays, Logan et al., 2016b; great-tailed grackles, Quiscalus mexicanus, Logan, 2016). Aesop’s Fable tasks require subjects to insert objects into water-filled tubes to obtain out-of-reach floating rewards.

In the corvids that have been tested using this object-dropping task so far, we see a common pattern, irrespective of whether they are habitual tool users. Namely, they are capable of learning the object-dropping task, but only once they have experienced an object falling into a tube, which usually occurs when they accidentally knock an object off the ledge into the tube. This finding suggests that the birds need to see the object fall, and once they have, they can learn to solve the rest of the task. This raises the question of whether they need direct experience of manipulating the objects and observing them fall into the tube or whether witnessing another individual’s solution to the problem will suffice in learning the task. So far, only two birds have solved the object-dropping task after observing a conspecific demonstrator: one rook (Bird & Emery, 2009b) and one New Caledonian crow (Mioduszewska, Auersperg & Von Bayern, 2015), though only the latter study aimed to explicitly test for influences of social information use on learning this task. New Caledonian crows are habitual tool users in the wild (Hunt, 1996), whilst rooks—like Eurasian jays—are not, though rooks have shown tool-use and manufacture proficiency in the lab (Bird & Emery, 2009b). Both rooks and crows are more social than jays in that rooks form large flocks for breeding, foraging and roosting, while New Caledonian crows tend to form extended family groups that are fairly tolerant of their neighbours (Goodwin, 1986; St Clair et al., 2015).

We also investigated whether Eurasian jays would choose the colour that was demonstrated to be rewarded in a two-choice colour discrimination test. Unlike the object-dropping task, this is a fairly simple task and corvids, including Eurasian jays, have been shown to be capable of making colour discriminations (ravens: Range, Bugnyar & Kotrschal, 2008; Eurasian jays: Clayton & Krebs, 1994; G Davidson, R Miller, E Loissel, L Cheke & N Clayton, 2016, unpublished data). Furthermore, this test has explicitly been used previously to demonstrate use of social information in other corvids, namely common ravens and carrion crows, where all the individuals that were tested chose the demonstrated colour (Miller, Schwab & Bugnyar, in press). Ravens and crows are social species with high fission–fusion dynamics, being highly social in the non-breeding season, and territorial in the breeding season (Goodwin, 1986). We conducted the task in a comparable manner to Miller, Schwab & Bugnyar (in press) to allow for direct comparison between these two corvid studies. The inclusion of both tasks in the present study allowed us to compare jay performances with social corvid species that have been shown to use social information on the same tasks. Furthermore, the use of both tasks enabled us to control for potential influences of task affordances, such as difficulty. Namely, even if the object-dropping task was too difficult to learn socially, we would still be able to detect whether the jays use social information in the simpler colour-discrimination task.

The general cognitive tool-kit hypothesis (Emery & Clayton, 2005) may predict that relatively asocial jays, like the more social New Caledonian crows, rooks, ravens and crows, would use the information provided by the demonstrator, as they may have retained the capacity to use social information (i.e., information made available by others). Alternatively, jays may differ from the more social corvids in their use of social information, as they may have secondarily lost this ability due to lack of selection pressure from an asocial environment.

Methods

Subjects

The subjects were 16 hand-reared juvenile Eurasian jays (eight females, eight males) hatched in May 2015. The birds were hand-reared as a group in 2015, and socially housed within a large outdoor aviary (9 × 16.5 × 6 m) at the Sub-department of Animal Behaviour in Madingley, Cambridge. Birds were sourced from wild nests at 10 days of age by a registered breeder under a Natural England License to NSC (20140062). The subjects consisted of five sibling groups (one pair, three groups of three birds, and one group of four birds), and one individual that had no siblings. Testing took place in indoor test compartments (2 × 2 × 1 m), with which the birds were familiar, as they were fed their daily diet within these compartments and had constant access to them outside of testing sessions. The birds could be separated individually, in pairs or sub-groups within these compartments as required. One female bird (‘Sjoika’) did not participate in either experiment, as she could not reliably be separated individually in the compartments. Subjects were identifiable using unique colour leg-ring combinations. Prior to and during testing, subjects had access to their daily diet, which consisted of soaked dog pellet and boiled vegetables, and water. Rewards for both experiments were live mealworms, which are a highly valued food item, reserved only for training and testing. Experiment 1 was conducted in October 2015 and Experiment 2 in November 2015.

Animal ethics

These experiments were conducted under approval from University of Cambridge Psychology Research Ethics Committee (application number: pre.2013.109) and the European Research Council Executive Agency Ethics Team (application: 339993-CAUSCOG-ERR).

Video summary

A video shows examples from both experiments: https://youtu.be/sU_5dPToxys. Experiment 1: trained group, Solving Task (Stuka); Experiment 1: observer group, Test Trial 5 (Gizmo); Experiment 2: observer group, Test Trial (Gizmo).

Experiment 1: Object-Dropping Task

Materials

The testing apparatus was a clear Perspex ‘object insertion’ apparatus (total height = 13 cm) consisting of a tube and a box (height = 10.5 cm, depth = 6.5 cm, width = 11 cm) containing a collapsible platform (based on the design in Bird & Emery, 2009b). Objects could be inserted into a tube (length = 8 cm, diameter = 5 cm), causing the collapsible platform at the bottom of the tube to release from a small magnet holding it in place. Once released from the magnet, a food reward was dispensed to the subject (Fig. 1). Several clear, plastic rings and one additional removable platform (length = 13 cm, width = 13 cm) that attached to the exterior of the tube were used for the earlier training stages. A blue ring was added to the top of the tube to increase the salience of this area. Only one object was required to drop the collapsible platform and release the reward. Spherical, black metal, hollow objects were used (measuring 2 cm in diameter and weighing 4–5 g; Fig. 1), with three thin pieces of black plastic string woven through the middle of each object and tied in a knot on each side to allow the birds to pick up the object more easily and prevent objects from rolling away.

Figure 1 Experiment 1 set up: stages of the object insertion apparatus.

(A) The removable platform at the top of the tube, (B) the removable platform at the bottom of the tube, and (C) the final stage apparatus (no removable platform). Photo: Rachael Miller.

Procedure

Subjects were separated into three groups: a trained group that had no prior experience with the apparatus and had never seen another solve it, but were trained to correctly solve it by the experimenter (three males, three females); an observer group that observed a trained conspecific solve the task (three males, three females); and a control group that received no training on the task and did not see any bird interact with the task (two males, one female).

Table 1 Stages of the object insertion apparatus.

Training stages: training the trained group, and subsequent to their tests, the observer and control groups, to insert objects into the tube to release the food reward. Training stages occurred in the following sequence: 1-2-3. Demonstrator stages: birds in the observer group watched the demonstrator solve the apparatus 40 times per stage before being presented with the final stage apparatus in a test trial. Demonstration stages occurred in the following sequence: 3-1-2-3.

Stage	Removable platform position	Object position	Figure 4 corresponding image	
	Training	Demonstration	Training	Demonstration		
1	Top of the tube	Top of the tube	Platform. Object baited with insect and then not baited	Table	a	
2	Gradually lowered down the tube using plastic rings until at the bottom of the tube	At the bottom of the tube	Platform or table	Table	b	
3	No platform	No platform	Apparatus base or table	Table	c	

Habituation and spontaneous object dropping

All subjects were habituated to the apparatus and the object separately by presenting them with small food rewards on top and beside the apparatus and object. Subjects were first presented with a baited object on the table until they retrieved the reward in five consecutive trials. They were then presented with the object insertion apparatus in the stages outlined in Table 1 and Fig. 1. Namely, the apparatus was presented in three scenarios to aid in learning how to correctly solve the apparatus: 1. The removable platform was placed at the top of the tube (Fig. 1A) to allow the object to be placed on the rim of the tube so the bird could easily accidentally knock the object into the tube by nudging it when attempting to obtain bait from under the object; 2. The removable platform was placed at the bottom of the tube (Fig. 1A) to encourage the bird to pick up the object and lift it up to the top of the tube to insert it; 3. The removable platform was removed (i.e., final stage apparatus; Fig. 1A) so the bird had to pick up the object from the table to insert it into the top of the tube. Rewards were placed on the apparatus, as well as underneath it, with the collapsible platform in the dropped position, until subjects retrieved all available rewards per trial in five consecutive trials.

All subjects were then given one five min test trial to determine whether they would spontaneously pick up and drop the object into the tube prior to being allocated to a group. During this test, the final stage apparatus (Fig. 1A) was presented to each subject with the object placed on the table beside the apparatus. No birds spontaneously solved the apparatus within the five minutes, therefore they were randomly assigned to one of three groups: trained, observer or control. Birds were allocated to groups by choosing names from a container: one ‘male only’ and one ‘female only’ container ensured a balanced sex ratio in each group (three males, three females for the trained and observer groups; two males, one female for the control group).

Trained group

We first trained birds in the ‘trained group’ to successfully solve the task by inserting objects from the table into the tube and obtaining the reward. We used the training stages outlined in Table 1 and Fig. 1 to gradually increase their proficiency from accidentally inserting baited objects balanced on the rim of the tube to nudging objects down the tube with the use of a removable platform attached to the outside of the tube (stages 1–2; Table 1), until they picked up objects from the table to insert into the tube without the removable platform present (stage 3; Table 1). In training stage 1, the object was baited with an insect on intermittent insertions for the first 1–2 training sessions (3–21 insertions, mean = 11 insertions). A session for the trained group lasted 5–10 min and was not restricted to a specific number of object insertions, but rather determined by the subject’s motivation and performance in that particular session. A maximum of two training sessions were run per day. An object insertion was considered proficient if it was nudged or dropped directly into the tube, as opposed to being knocked in accidentally by removing the baited insect, or first pushing it around on the platform or dropping it onto the table from the platform.

Subjects moved from stage one to stage two when they had accidentally knocked the object into the tube on 10 consecutive insertions (Fig. 1A). The removable platform was then gradually moved down the tube during stage two until the subject inserted the object from the platform when it was placed at the bottom of the tube on 10 consecutive insertions (Fig. 1B). If subjects struggled with progression to the next stage (e.g., stopped inserting the object), they returned to the previous stage, with the aim for each training session to ‘end on a high’ (i.e., with a reward for inserting the object). A bird was considered to have solved the task when they had inserted the object from the table into the final stage apparatus and obtained the reward in 10 consecutive insertions (Fig. 1C).

We then selected one bird from the trained group (Homer) to demonstrate how to solve the apparatus to the observer group. This bird was selected to be the demonstrator because he was motivated and reliable during training (e.g., he was easy to call into the test compartments and comfortable being close to humans), and solved the task during training fairly quickly. Homer was 100% accurate when he demonstrated for observers; therefore observers never saw failed attempts.

Observer group

Observers saw the demonstrator successfully solve the apparatus 40 times per stage, using the following stage order: 3-1-2-3 (i.e., observers saw 40 demonstrations of stage three, then 40 demonstrations of stage one, etc.; Table 1). This resulted in a total of 160 observations of successful solves per observer bird. Observers were given four demonstration sessions of 10 solves per session per stage. The stages were the same as those used for the trained group (Table 1; Fig. 1). As these stages facilitated the training of the trained group to solve the task, we might expect that aspects of these stages are helpful for learning the task, hence including demonstrations of each stage. Each demonstration session lasted approx. three min, with a maximum of two sessions run per day. The demonstrations took place on a table in one compartment, with the observers located in an adjacent, but separate compartment with free visual access between compartments via mesh panels. There were three to four observer birds per adjacent compartment and there were sufficient perches for all observers to view the demonstrations at the same time. The observer group was split into two smaller sub-groups of three birds per group for observations to ensure each bird had sufficient visual access of the demonstrator and to reduce crowding within the test compartments. Each observer subject had the opportunity to watch 16 demonstration sessions, with one or two sessions per day, ensuring that each observer had ample opportunities to observe demonstrations.

Immediately after an observer saw 40 demonstrated solutions at a particular stage, the observer was visually isolated and presented with the object insertion apparatus at the final stage (i.e., no removable platform and with the object on the table). They were then given one five-min test trial to determine whether they had learnt to solve the task. Observer subjects received five five-min test trials: one pre-demonstration test trial that all birds received to determine whether they spontaneously solve the task, and observer birds received four test trials immediately after observing demonstrations at each stage (stages 3-1-2-3; Table 1). Each test trial therefore took place on a separate day, over a period of 15 days. During all test trials, the observer subject was presented with the final stage apparatus with the object on the table. To solve the task, the subject was required to pick up the object from the table and insert it into the tube to release the collapsible platform and obtain the reward. The longest time that any subject waited between observing the last demonstration session of each stage and their own test trial was 10 min.

Control group

The control group did not receive any object insertion apparatus training or demonstrations, and were presented with the ‘final stage’ object insertion apparatus the same number of times that the observer group received the apparatus (i.e., five test trials). Test trials were run on the same test days as the observer group to avoid any potential differences between the groups due to age or other environmental factors.

Data analysis

All training and demonstration sessions and test trials were videotaped, as well as being live coded. We recorded the number of (accidental and proficient) insertions required for the trained group individuals to complete each training stage and solve the task (i.e., to insert an object from the table into the tube at the final apparatus stage in 10 consecutive insertions). For the observer and control groups, we recorded whether the subject solved the task (i.e., inserted an object from the table into the tube at the final apparatus stage, and interacted with the apparatus or object).

To determine whether individuals in the observer group interacted with the apparatus and object more than individuals in the control group during tests, we conducted a generalised linear model (GLM) using a Poisson distribution with a log link in R v3.2.1 (function: glm; R Development Core Team, 2015). We combined the total number of times a bird touched the apparatus and object per trial (response variable) to examine whether it varied by trial number or group (control or observer; explanatory variables). We conducted a generalised linear mixed model (GLMM) using a Poisson distribution with a log link (R package: lmerTest, function: glmer, Kuznetsova, Brockhoff & Christensen, 2015) to determine whether the observer group interacted more with particular parts of the apparatus or object after having seen the demonstrator solve the task. We examined whether the number of touches (response variable) varied according to the location that was touched (apparatus base, apparatus tube, or object) by group (control or observer; explanatory variables) with bird ID as a random effect. To examine whether observer jays touched the apparatus/object sooner than control jays, we conducted the same GLMM just mentioned, but with a different response variable: the latency (in seconds) to touch the apparatus or object per test trial per bird.

To examine the level of certainty associated with each model, the respective models were compared with all model combinations and their Akaike weights, which sum to one across the models, evaluated (R package: MuMIn, function: dredge; Bates, Maechler & Bolker, 2011). A model was considered highly likely given the data if it had a high Akaike weight (>0.89) relative to the other models (Burnham & Anderson, 2002).

Once Experiment 1 had been conducted, all of the birds in the control and observer groups were trained to insert objects into the object insertion apparatus. We recorded the number of (accidental and proficient) insertions required for the observer and control groups to complete each training stage and solve the task. We examined whether birds in the observer group solved the task faster than birds in the trained or control groups using a GLM in R. The number of object insertions required to complete stage three (insert the object from the table into the tube in 10 consecutive insertions; response variable) was compared across conditions (trained, observer, control; explanatory variable) using a Poisson family with a log link.

Results

None of the jays solved the task spontaneously in the initial trial (i.e., prior to any training, demonstrations or frequent exposure to the apparatus). In the trained group, all six jays learned to drop objects over a period of eight to 21 training sessions (4–11 days). In the observer group, zero of six jays learned to drop objects by observing the demonstrator. In the control group, zero of three jays learned to drop objects without training or demonstrations. Only one bird (Gizmo—observer bird), on her final test trial, lifted the object high up while standing near the tube, but she did not insert it into the tube.

All observer and control subjects generally interacted with the apparatus and/or object during test trials (in 44 of 45 test trials; with the apparatus in 39 trials and the object in 34 trials). Individuals in the observer group did not touch the apparatus or object more frequently than individuals in the control group (mean touches = 11 and 9, respectively; Table 2: Model 1). The Akaike weight for this model was very low (0.11), and it was the third ranked model, indicating a high level of uncertainty, therefore it is likely that there was not enough data for the model to draw strong conclusions, or the effects were too small to detect.

Table 2 Did observers learn what to attend to from the demonstrator?

Results from the GLM (Model 1) and GLMM (Model 2) examining whether individuals in the observer group touched the apparatus and object more frequently than control individuals (Model 1) or whether they interacted more with particular parts of the apparatus (base or tube) or object (Model 2). Model 3 (GLMM) examined latencies to first touch per trial to determine whether individuals in the observer group first touched the apparatus/object sooner than control birds. SE: standard error, z:z value, p:p value, the rows in italics list the variance and standard deviation of the random effect.

Model	Variable	Estimate	SE	z	p	
1	Intercept (controls)	3.19	0.17	18.42	<0.001	
	Trial	−0.37	0.07	−5.62	<0.001	
	Observers	−0.17	0.21	−0.83	0.41	
	Trial*Observers	0.16	0.08	2.06	0.04	
2	Intercept (apparatus base, controls)	1.19	0.25	4.83	<0.001	
	Object	−0.25	0.20	−1.12	0.23	
	Tube	−0.32	0.21	−1.54	0.12	
	Observers	0.44	0.29	1.50	0.13	
	Observers*object	−0.37	0.24	−1.51	0.13	
	Observers*tube	−0.14	0.24	−0.59	0.56	
	Bird ID	0.12	0.35			
3	Intercept (controls)	4.32	0.21	20.88	<0.001	
	Observers	−1.22	0.26	−4.78	<0.001	
	Bird ID	0.13	0.35			

While the number of interactions decreased with increasing trial number in control individuals, there is weak evidence that observer individuals had relatively more interactions with the apparatus and object in later trials than control individuals (Table 2: Model 1). There was only weak evidence because the Akaike weight for the top-ranked model, which was the full model, was only 0.46, indicating that there was a high degree of uncertainty in this model. There was no evidence that birds in the observer group interacted more with particular parts of the apparatus or object after seeing the demonstrator solve the task compared with control birds (mean touches = 4 and 3, respectively; Table 2: Model 2). When comparing the latency to the first touch between control and observer groups, observer birds touched the apparatus/object significantly sooner than control birds (mean = 23 and 83 s, respectively; Table 2, Model 3; Fig. 2). This model was highly likely given the data because its Akaike weight was 0.99. The data in Fig. 2 shows that there was no initial difference in latencies between control and observer groups during their spontaneous test trial (trial 1), which was before the observer group had access to social information about the apparatus. The difference between the two groups occurred in trials 2–5 where, after the social demonstrations, observer latencies stayed the same, while the control group’s latencies increased.

Figure 2 Experiment 1: object-dropping test trials for observer and control groups.

Mean latency to first touch of the apparatus or object per trial for Observer (white boxplot) and Control (hatched boxplot) groups.

Following this experiment, all nine jays in the observer and control groups underwent training to drop objects over a period of 8–12 training sessions (five to seven days). Therefore, the number of object insertions required to reach proficiency was compared between the trained, observer, and control groups. Birds in the trained group required more insertions to solve the task (i.e., to insert objects from the table into the tube of the final stage apparatus; mean insertions to solve = 167, GLM estimate = 0.39, SE = 0.06, z = 6.26, p < 0.001), than observer and control birds. Birds in the observer (mean insertions to solve = 114, GLM estimate = 0.01, SE = 0.07, z = 0.20, p = 0.84) and control (mean insertions to solve = 113, GLM [intercept] estimate = 4.72, SE = 0.05, z = 86.86, p < 0.001) groups did not differ in the number of insertions (Fig. 3; ESM1 Table S1).

Figure 3 Experiment 1: number of object insertions to solve.

Total number of object insertions to solve the object-dropping task per group.

Experiment 2: Two-Choice Colour Discrimination Task

Materials

This set up consisted of two plastic cups—one black and one white (diameter = 6 cm, height = 14.5 cm). Cups were spaced 30 cm apart on a wooden board (50 cm × 15 cm). Each cup was attached to its own metal rod so they could move up and down independently, but they were prevented from being removed entirely from the rod by a bolt. Cups could be lifted upwards to reveal a hidden reward (Fig. 4). Two live mealworms were placed underneath each cup.

Figure 4 Experiment 2 set up.

Two-choice colour discrimination task where observers only saw a demonstrator find food under the white cup. Photo: Sarah Jelbert.

Procedure

Demonstrator training

One bird acted as a demonstrator—Homer—the same demonstrator as in Experiment 1. In visual isolation from the observer group, Homer received four sessions (5–10 min per session) of 10 trials per session, where only one cup—the white cup—was baited (‘demonstrated’ cup) and the other cup—the black cup—was locked down using the bolt so it could not be lifted. To pass demonstrator training, Homer had to consistently lift only the demonstrated cup in all 10 consecutive trials within a session and not touch or try to lift the other cup before he could move on to the demonstrations for observers. Homer touched both cups in session one and two, but passed criterion in session three. He was given four training sessions in total to ensure comparability with the number of demonstrator training sessions used for the carrion crows and ravens in Miller, Schwab & Bugnyar (in press). Homer chose the white cup 100% of the time during demonstrators for observers; therefore observers did not see any incorrect choices.

Demonstrations for observers

The observer group consisted of seven birds (four females and three males) in order to be comparable with the sample size in Miller and colleagues (2016). These individuals also participated in Experiment 1: three from the trained group, three from the observer group, and one from the control group. In an adjacent compartment with visual access to the observers, the demonstrator lifted the demonstrated cup (white) and obtained the reward in four sessions, with 10 trials per session. Both cups were baited and could potentially be lifted, though the demonstrator only lifted the demonstrated cup. The demonstrated cup location (left or right) was counterbalanced across trials. Each observer watched one session per day.

Testing observers

After observers had seen Homer lifting the demonstrated cup 40 times, they were tested in visual isolation from the group. Each observer was presented with the cups, both cups were baited out-of-sight of the observer and we recorded which cup they touched first. They were given one test trial, which lasted up to three minutes (all subjects interacted with the cups within three minutes). They were allowed to touch both cups. The location of the demonstrated cup was randomized across subjects. If they touched the demonstrated cup (white) first, we considered this to be using social information from the demonstrator.

Data analysis

We recorded the colour and latency of the cup first touched by the demonstrator during training and demonstration trials, and by the observers during their test trial. The data were analysed using SPSS version 21 for the exact two-tailed Binomial tests, and R for the t-test. RM and KL both coded 20% of all videos across both experiments, with KL acting as a naïve coder, and inter-observer reliability was excellent (Cohen’s kappa k = 0.989, p < 0.001).

Results

Jays did not choose the demonstrated colour above chance levels (Binomial test: p = 0.453). Two of seven jays (one male, one female) chose the same coloured cup (white) as the demonstrator (i.e., copied the demonstrator), while the other five jays (three females, two males) chose the non-demonstrated cup colour (black; Table 3). In comparison, Miller, Schwab & Bugnyar (in press) found that eight of eight crows (five females, three males) and eight of eight ravens (three females, five males) copied the conspecific demonstrator, which was significant (Binomial test: p = 0.008 for each species). We additionally examined whether there was a difference in the latency to make the first choice between the birds that chose the demonstrated colour versus those that did not. The jays that chose the demonstrated colour did not have shorter latencies to their first choice (Welch two-sample t-test: t = 0.88, p = 0.47, n = 7, 95% confidence interval = − 36–57; data in ESM1 Table S1). We also explored whether relatedness influenced likelihood to copy the demonstrator. Zero of two jays that selected the demonstrated coloured cup (Binomial test: p = 0.5, n = 2) and two of five jays that did not select the demonstrated coloured cup were siblings of the demonstrator bird (Binomial test: p = 1.00, n = 5). The birds did not appear to show a group side bias because they did not select the cup on the same side regardless of colour (Table 3: Binomial test: p = 1.00, n = 7).

Table 3 Two-choice colour discrimination task results.

The birds observed the trained demonstrator Homer lifting the white cup to retrieve a mealworm on 40 consecutive trials.

ID	Sex	Demonstrated colour	Chosen colour (first choice)	Location of chosen colour	Latency to first choice (s)	
Dolci	F	White	Black	Left	19	
Stuka	F	White	Black	Right	51	
Horatio	M	White	White	Left	44	
Booster	M	White	Black	Left	20	
Lintie	F	White	Black	Right	12	
Gizmo	F	White	White	Right	25	
Roland	M	White	Black	Left	19	

Discussion

We found that relatively asocial Eurasian jays did not use social information (i.e., information made available by a conspecific) in the form of copying the choices of others in either task. In Experiment 1 (object-dropping task), birds in the observer group first touched the apparatus and object significantly sooner than birds in the control group, indicating a form of social learning called stimulus enhancement. Stimulus enhancement attracts the attention of an observer towards a specific object where the model acts (Giraldeau, 1997). However, observing a conspecific demonstrator did not facilitate solving the object-dropping task in Experiment 1, or result in colour choice copying in Experiment 2.

Although corvids, including Eurasian jays, can be trained in the object-dropping task, it is possible that this task is too difficult for social learning to facilitate the solution, except for the occasional individual (i.e., one rook: Bird & Emery, 2009b; and one New Caledonian crow: Mioduszewska, Auersperg & Von Bayern, 2015), thus masking whether Eurasian jays are able to use social information by observing, and thus learning from, a demonstrator. In the present experiment, solving this task required the observer birds to copy several actions of the demonstrator: lifting the object from the table to insert it into the tube to drop the collapsible platform and obtain the reward, rather than just knocking an object into the tube from the tube ledge, which is typically stage one of training. In further support of the suggestion that this task is difficult for birds to learn is that only one bird has spontaneously solved the object-dropping task in a previous study (one New Caledonian crow; Mioduszewska, Auersperg & Von Bayern, 2015), without any demonstrations or training. Additionally, birds, including corvids, typically require a relatively large number of training trials to learn to solve this task, indicating that it is fairly difficult to learn even with explicit training (e.g., 90–275 trials in the present experiment; 135–362 trials in great-tailed grackles: Logan, 2016; 76–255 trials in California scrub-jays: Logan et al., 2016b—though note that definitions for reaching proficiency differ between these studies and the current experiment).

It is therefore possible that the jays obtained some information from the demonstrator, but potentially this information was not sufficient to enable them to complete the task (i.e., to insert the object from the table into the tube). Therefore, we assessed whether there was any evidence that the jays attended to the demonstrator, despite not being able to solve the task following the demonstrations, by measuring differences in the number of interactions with the apparatus and object between the control and observer groups. Individuals in the observer group were not more likely to touch the apparatus or object than individuals in the control group. Observer individuals touched the apparatus and object in later trials more than control individuals, indicating that jays may have been more persistent after having seen another bird solve the task. However, it should be noted that the models showed only weak evidence for these two findings.

We also found that the observer group solved the object-dropping task significantly more quickly than the trained group; however, there was no difference in the rate of learning (i.e., total number of insertions required to solve the task) between the observer and control groups. The strongest evidence of any form of social learning was in the form of stimulus enhancement: observer birds that had seen a demonstrator interact with the apparatus and object first touched these elements significantly sooner than control birds that had never observed another touching the apparatus. It is possible that increased exposure to the apparatus may have facilitated learning in both the observer and control groups, perhaps by removing neophobia of the apparatus (although all birds were habituated to the apparatus prior to testing), and/or some social facilitation of attraction or attention to the apparatus, as opposed to learning the actions to perform the task. However, it is unclear whether observers attended to social information provided by the conspecific or whether they would have learned about the task by observing a ‘ghost control’ where the object was inserted into the tube in the absence of a conspecific. Future research incorporating ghost controls could distinguish between whether jays attend to social information about what to attend to or whether they solely attend to the relevant object movements and reward outcomes.

In Experiment 2, in comparison with the object-dropping task, the colour discrimination task was relatively simple as corvids are capable of making colour discriminations (Clayton & Krebs, 1994; Range, Bugnyar & Kotrschal, 2008). For example, there is evidence that juvenile Eurasian jays can discriminate between colours in similar two-choice discrimination tasks. Davidson and colleagues (G Davidson, R Miller, E Loissel, L Cheke & N Clayton, 2016, unpublished data) trained half of a group of Eurasian jays to associate a yellow coloured object with a reward and a green coloured object with no reward, and the other half to associate the green object with a reward and the yellow object with no reward. The jays then demonstrated proficiency by flying to the perch where the rewarded colour was located.

Further, the same task used in Experiment 2 was used previously in eight ravens and eight carrion crows, and all birds chose the demonstrated colour (Miller, Schwab & Bugnyar, in press). While the methods have some limitations (e.g., no counterbalancing of rewarded cup colour, using only one demonstrator whose characteristics might have made him less likely for observers to attend to, low statistical power from only one trial per bird), we ran this task in a comparable manner to Miller, Schwab & Bugnyar (in press) to allow for direct comparison between these two experiments, including the use of one male who was a same-age conspecific demonstrator to an observer group and one test trial. Additionally, all birds were hand-reared in species groups in a similar manner, tested by the same experimenter (RM) and similar sample sizes were used (eight ravens, eight crows, seven jays). We also similarly controlled for the influence of spatial location by randomising the location of the demonstrated cup across subjects, and we found no group-level bias for one location (right/left) over the other (Table 3).

There were two notable differences between these experiments. Firstly, the colour discrimination task used different colours: blue and yellow cups in Miller, Schwab & Bugnyar (in press) compared with white and black cups in the present experiment. The justification for this difference was the need to avoid a possible overlap between this experiment and the prior experience of the jays with several different colours in differing reward scenarios during previous studies (e.g., G Davidson, R Miller, E Loissel, L Cheke & N Clayton, 2016, unpublished data). Furthermore, Shaw and colleagues (2015) suggest that colour discrimination tasks should aim to use gray scale cues (e.g., light vs. dark gray) to avoid innate species-level colour preferences. We cannot entirely rule out innate colour preferences because we did not transfer birds to novel colour combinations. However, innate preferences would likely have been expressed at the species level, which did not occur here because jays randomly chose white and black cups in their first trials.

Secondly, the jays were juveniles, whereas the ravens and crows were sub-adults. Therefore, it is possible that social learning in the jays may not have developed by this early stage. However, this is unlikely given that juveniles in other relatively asocial species exhibited social learning whereas adults did not (Lupfer, Frieman & Coonfield, 2003; Noble, Byrne & Whiting, 2014). To our knowledge, no corvid studies have compared juvenile and adult social information use. However, object permanence in Eurasian jays, which relates to caching development, develops at a similar stage as in other corvids (ravens: Bugnyar, Stowe & Heinrich, 2007; California scrub-jays: Salwiczek et al., 2009). Specifically, jays reach a full (i.e., stage six Piagetian) understanding of object permanence within their first few months of life (Zucca, Milos & Vallortigara, 2007). As the jays we tested were more than a few months of age, we do not expect their behaviour to differ from adult behaviour with regard to social learning. The finding that the jays behaved differently from the more social carrion crows and ravens in the use of social information in this task is important. It raises the question of whether these more social species—as with the more social rook (Bird & Emery, 2009b) and New Caledonian crow (Mioduszewska, Auersperg & Von Bayern, 2015)—might be able to learn to copy the demonstrator in the object-dropping task (Experiment 1).

Previous experiments have indicated that Eurasian jays do attend to social context in caching and mate provisioning (Shaw & Clayton, 2012; Shaw & Clayton, 2013; Ostojić et al., 2013; Shaw & Clayton, 2014; Ostojić et al., 2014; Legg, Ostojić & Clayton, 2016). It is therefore still possible that jays use social information, but not for copying others’ choices, as none of the previous experiments required the birds to copy a demonstrator. Jays may also be more likely to pay attention to and copy different demonstrators, such as older, more affiliated or related individuals, as model identity has been found to influence social learning in other corvids (ravens, jackdaws: Schwab, Bugnyar & Kotrschal, 2008a; Schwab et al., 2008b). For example, the presence of siblings enhances social learning in ravens (Schwab et al., 2008b). Our demonstrator was a sibling of some of the observers, which suggests that there was no influence of relatedness to demonstrator on likelihood of copying in Experiment 2. However, our experiment was not designed to test the relationship between relatedness and social learning and we do not have the statistical power to make a firm conclusion on this point.

The use of social information is a process with several stages, which are likely to be sequential and distinct: acquisition (observing another), application (performing the observed behaviour, not necessarily successfully) and exploitation (successfully performing the observed behaviour in a way that gives the individual an advantage; Carter, Tico & Cowlishaw, 2016; Guillette, Scott & Healy, 2016). For instance, in chacma baboons (Papio ursinus), the average individual acquired social information on <25% of occasions and exploited social information on <5% of occasions, and information use was dependent on phenotypic constraints such as network position and dominance status (Carter, Tico & Cowlishaw, 2016). The results of Experiments 1 and 2 demonstrated that Eurasian jays did not appear to apply or exploit the social information available even though they had the opportunity to acquire it. Although we reiterate that social species also do not show a strong capacity to socially learn the object-dropping task in Experiment 1.

In conclusion, Eurasian jays did not appear to use social information in the form of copying the decisions of a conspecific in the object-dropping and colour discrimination tasks, which vary in difficulty. However, their attention was drawn to the apparatus and object in the object-dropping task as indicated by observers touching these components sooner than control birds. In previous studies with social corvids, the birds had been explicitly tested for influences of social information on learning the object-dropping task in only one study, with only one New Caledonian crow learning the task following a conspecific demonstration (Mioduszewska, Auersperg & Von Bayern, 2015). We also know that, when tested using very similar procedures, including the same lead experimenter, ravens and crows use social information in the colour discrimination task, in contrast to the jays. These corvid species vary in sociality, but all are more social than the jays. Our results from relatively asocial Eurasian jays are therefore consistent with those from relatively asocial Clark’s nutcrackers (Bednekoff & Balda, 1996; Templeton, Kamil & Balda, 1999) in that social and relatively asocial corvids appear to differ in their use of social information with regard to copying the choices of others. The present experiment may indicate that Eurasian jays secondarily lost the ability to copy social information provided by a conspecific, at least in some contexts, while maintaining the ability to attend to the general movements of others, due to a lack of selection pressure from an asocial environment. However, more comparisons between social and relatively asocial corvids are needed to confirm this hypothesis.

Supplemental Information

Data S1 Raw data for the number of object insertions needed to solve the task

The number of object insertions (accidental and proficient) required per subject to solve the task (i.e., complete stage 3). The observer and control groups were trained following completion of Experiment 1.

Click here for additional data file.

We thank Elsa Loissel and Natalie Williams for their help in the early stages of preparing for Experiment 1 and for discussions. Thank you to Maggie Dinsdale, Sam Melvin, Sarah Manley, Ivan Vakrilov for animal care, to Ian Millar for help in apparatus construction, and to Mark Ghobain for assistance in hand-rearing the Eurasian jays. We are grateful to Jennifer Vonk, Jorg Massen, and Dawson Clary for manuscript feedback.

Additional Information and Declarations

Competing Interests

Author Contributions

Animal Ethics

Data Availability

The authors declare there are no competing interests.

Rachael Miller conceived and designed the experiments, performed the experiments, analyzed the data, contributed reagents/materials/analysis tools, wrote the paper, prepared figures and/or tables, reviewed drafts of the paper, coded videos.

Corina J. Logan analyzed the data, wrote the paper, prepared figures and/or tables, reviewed drafts of the paper.

Katherine Lister reviewed drafts of the paper, coded videos.

Nicola S. Clayton reviewed drafts of the paper, discussed the design and running of the experiments and was responsible for the overall maintenance of the corvid lab and research conducted therein.

The following information was supplied relating to ethical approvals (i.e., approving body and any reference numbers):

The study was conducted under approval from University of Cambridge Psychology Research Ethics Committee (application number: pre.2013.109) and the European Research Council Executive Agency Ethics Team (application: 339993-CAUSCOG-ERR).

The following information was supplied regarding data availability:

Data and R code for the social learning GLMMs is available at the KNB Data Repository at: https://knb.ecoinformatics.org/#view/doi:10.5063/F1PN93J1 (Miller & Logan, 2016).

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
