# Peer review of "Eurasian jays do not copy the choices of conspecifics, but they do show evidence of stimulus enhancement"

_PeerJ, doi:10.7717/peerj.2746_

## Round 0.1 · original submission · Major Revisions

First, let me say that I am very happy to see more explicit tests of the social intelligence hypothesis comparing performance of less social birds to that of social birds. I think this work is important and the data could make a significant impact on the field. I’ve now received two very thorough and helpful reviews of your MS. Both reviewers value the aims of the study and think that the data should be published. However, before the MS can be accepted, it requires significant revision regarding the framing and interpretations of the data. Both reviewers feel that the studies could benefit from a stronger rationale and framing to link the experiments, which could be assisted by including results and discussion for Exp. 1 before moving on to presentation of Exp. 2 and I agree. In addition, the introduction should contain more of a discussion on why sociality should predict better social learning. If learning from observation is a general ability, it might not matter whether information is conveyed socially or non-socially, for example.

In the fifth line of the abstract, get rid of “or not”. When using “whether”, “or not” is redundant and unnecessary. See lines 93, 304, 426 as well.

Use commas consistently after i.e. and e.g.

There should be a; before “however,” or begin a new sentence.

The sentence on lines 46-47 doesn’t seem to follow from what precedes it.

What kind of selection pressure would arise from an asocial environment that would cause a species to lose the ability to learn via social information? (lines 52-55)

On line 65, move the “only” to before “one test”. Similarly, on line 252, move “only” to after “lift”.

I’d hesitate to refer to Eurasian jays as solitary or asocial if they spend significant periods of time defending territories within mated pairs. You might instead stress that they do not live in large, social groups. But that isn’t quite comparable to being solitary, particularly if they do exhibit dominance hierarchies in caching contexts. Furthermore, your own test subjects are socially housed so you might differentiate between birds that have evolved to live relatively asocially compared to current housing conditions. Please provide even more context for comparing different species in the introduction.
You should also reference work showing that other less social species can learn via social information. For example Anna Wilkinson has conducted some interesting research on social learning in tortoises and other less social reptiles that should be cited.
It seems to me that a better control for Exp. 1 would be to have the object drop it into the tube via some mechanical function (in the absence of a live agent) to tease apart whether it’s really about social learning or just about seeing the relevant outcome/actions of objects that facilitates learning (like a ghost control). Although you cannot redo the experiment with the inclusion of such a control, you might note that your current procedures do not allow you to tease apart explanations for the mechanism underlying successful copying with a demonstrator. A ghost control should be suggested for future studies.

I agree with Reviewer 2 that it is difficult to use first trial performance of only 7 birds to demonstrate social learning in Exp. 2. Birds could have an innate preference for white cups, and half the birds could choose white on the first trial by chance alone, making it nearly impossible to obtain a statistically significant result that is meaningful. Done properly, the experiment would have counterbalanced the color of the correct cup across the observer birds and also tested for transfer using novel color combinations. Even if you are following the procedures used previously to test social birds, you must acknowledge the limitations of your approach.

I disagree with Reviewer 1 about the statistics. If you are using multiple observations from individuals, then the GLMM procedure is more appropriate than traditional ANOVAs. I do agree that information from the analyses could be included in the text, making the tables unnecessary.

·

Basic reporting

This is an interesting study on the use of social information by a non-social corvid species. In general, I think the experimental design is solid and the results are clear, and I would warrant publication. However, I do have some major concerns with some of the conclusions drawn and some of the methods used, that in my opinion do need revision.

A general comment about the structure: Maybe it is nice to add the results of Experiment 1 immediately after the methods of Experiment 1, and then continue with Methods Experiment 2....Results Experiment 2

Experimental design

I have 3 problems with the experimental design:

First, your testing juvenile birds only. These might be developmentally constraint regarding social learning. Whereas there is nothing you can do about that now, you should discuss this in the discussion. You could in that discussion, however, also link it to the raven and crow data, that if I'm correct also where juveniles, and who nonetheless showed discrimination, and thus discard my argument. Anyway, it needs to be discussed.

Second, you use only one individual as demonstrator and in experiment 2 also only 1 colour. This creates pseudoreplications that may influence your data: e.g. all birds dislike Homer and do not pay attention to him, and consequently do not learn socially from him, and not because they cannot. Similarly, the birds may have a preference for black that is not so easily 'unlearned' socially. Again something that cannot be changed anymore, yet that needs to be discussed.

Finally, the analyses are sometimes unnecessarily complicated, and I see no justification for the models that are run: e.g. why do you run a model rather than a 2-way anova on line 289ff? And if you run a model, why did you choose a Poisson distribution. Did your data follow that distribution?
Similarly, line 305ff. This could be analyzed by a simple Anova or a Kruskal Wallis test (if your data are non-parametric), and again you provide no justification for using a Poisson distribution.
I do not think this will change the results, though I think that using simple stats would clarify your results even further.

Additionally, why did the n drop from 15 to 7 birds in experiments 1 and 2 respectively?

Validity of the findings

Even giving my problems with the experimental procedure (which should be discussed) I do think the results are robust and clear. However, I think the final conclusion is not justified; i.e. that social information use is secondarily lost in the jays. And my problem with this already starts with the assumptions made in the introduction. First there is the hypothesis that all corvids have a general cognitive tool-box that was already present in the common ancestor. Second the common ancestor of corvids is hypothesized to have been social. Given those two assumptions it is then assumed that this common ancestor should have had specific social cognition, like the use of social information, and that when current a corvid species does not have that anymore, this is lost.

This is based on multiple assumptions that can just as easily be false; i.e. maybe even though the common ancestor was social, it did not yet have sophisticated social cognition, and this only later evolved in the more social corvid species (note that the Jays are philogenetically an out-group in the comparisons made in this paper).

To make such claims you need a phylogenetically controlled study on multiple species that compares the amount of changes when considering that the common ancestor already had a trait or not, and then take the most parsimonious explanation. Given that you compare at most 3 species, such an analyses is not possible.

You can off course speculate about it, but then please present it as such and also present alternative explanations.

Additional comments

Abstract: the first and second sentence don't really follow. Maybe add socio- to ecology.

Also in the second sentence, maybe add such:....because only one SUCH test

Introduction:
l42. You say some corvids, yet then come with examples of only one. Maybe add some references to similar studies in other corvids.

l.46-47. I do not understand how it suggests that.

l. 47ff. Later on you give examples of corvids having that ability, so it is not completely unknown.

l. 50ff & l. 136. Or alternatively it was not present in the common ancestor and has evolved in social corvid species only.

l. 72. Omit 'so far' as you already mention 'at present'

l. 84-85. ?? What do you mean with different roles?

Methods:
l. 155 I assume Experiment 2 was conducted in November 2015 and not 2016

l. 164, they were used for what? Maybe exchange this and the following sentence

Results:
l. 346. Maybe first explain what kind of comparison you are doing here before giving the result.

Discussion:
l. 468. as mentioned before, also provide alternative explanations.

·

Basic reporting

The authors examined whether Eurasian jays will use social information to aid performance in two tasks. The first task involved observing a demonstrator lift an object and drop it in a tube apparatus resulting in a food reward being released. The second task required jays to select from one of two food cups after watching a demonstrator retrieve food from a particular cup. The birds showed no evidence that the experience of watching a demonstrator successfully complete the task aided their own performance.

The study is interesting and there is remarkably little research into whether social learning is an adaptive specialization of social living, so the authors are on fertile ground in that respect. My main critique of the manuscript is that the writing lacked precision, particularly in the methods, but throughout the rest of the manuscript as well.

1. Both the abstract and the introduction, at times, gave the impression that a social species would also be included in this study (e.g., lines 17-18, 74-75). This was not the case, which really diminishes any conclusions of whether social living affects social learning. The authors will need to carefully word these sections to avoid this potential confusion, avoiding statements implying a direct comparison of social and less social species.

2. Given that the primary motivation of the paper is to address whether social living results in the adaptive specialization of social learning, I would like the authors to provide more clear rationale for why they chose the two tasks that were conducted. The first task seems to have no potential for addressing this issue, as the evidence that social species use observation to solve this task is quite limited. The second task has more potential in this regard given the ‘successful’ results from social species. Typically, in multi-experiment papers, one experiment logically leads to the next, and that logic is described, but I don’t see that obvious link here. Did the authors want to vary task difficulty? If so, what were the a priori predictions for a less social species in different task difficulties? The authors should communicate this link more explicitly.

3. For the most part, the literature covered was appropriate. However, the authors need to provide a clear and precise definition of what they are considering ‘social information use’ – a very broad term. I assume the authors are trying to focus on whether Eurasian jays will copy the choices of others, but under the current descriptions provided, I see no reason to think that there is a difference between social and less social species – clearly both use social information in caching contexts (e.g., Clary & Kelly, 2011; Shaw & Clayton, 2012, Tornick et al., 2016). The authors could use the large modeling literature on social learning as the basis of a more refined definition of what they are studying (e.g., Laland, 2004, Animal Learning & Behavior). Furthermore, if focusing on copying the choices of others, then it would be worth mentioning reports of observational spatial memory, which both social and less social species demonstrate (Bednekoff & Balda, 1996), and could be considered copying the choice (i.e., copying the cache/search location) of a demonstrator. This would provide more balance to this section, in which the authors cite ‘social information use’ from a variety of different contexts for social species.

4. The number of figures and tables seems excessive for the amount of information that needs to be conveyed. Some of the tables simply contain results that should be reported in the text, whereas others contain redundant information with the figures and could be eliminated or combined (see specific comments).

Experimental design

1. The methods section lacked some precision and details needed to evaluate the experiments and, ultimately, replicate them. Thus, I largely have a variety of specific questions that can be found in the specific comments.

2. My only major criticism of the methodology is the choice to only conduct a single trial for the second task, rather than following the approach of Templeton et al. (1999). The problem with conducting only a single trial and analyzing with a binomial test is that it is seemingly a risky way of designing an experiment and analysis. With such a small sample, it would require all birds (7/7) copying the demonstrator to achieve statistically meaningful results. There are a variety of reasons why the birds may not have copied the demonstrator on the first trial. One explanation could be that less social species are just as capable of using social information, however they first sample the choices to determine the reliability of the demonstrator’s choices. It would be more informative to examine whether observers have steeper learning curves than those solving the task individually. Could the authors provide justification for only conducting a single trial?

Validity of the findings

1. I have a few specific questions regarding the analyses
a. Why were unequal groups created for Experiment 1? There were 15 birds, so I would expect 5 per group. Did the authors account for this imbalance in the statistical tests? Was variance examined to ensure it was roughly equal across groups?
b. Why was a GLM used for one analysis and GLMM for the others? How did the authors account for dependencies in the scores of the GLM? Was a random effect specified?
c. There does not seem to be any text in the results section devoted to the analysis described on lines 290-295.

2. The results section should be re-worked to include more statistical results in the text (rather than placing simple analyses into tables). The authors should also add in descriptive statistics for the tests, allowing the reader to better evaluate this section and the validity of the authors’ conclusions.

3. Did the authors look at latency to approach the apparatus (Exp. 1) or the cups (Exp. 2)? This could be used to further examine whether the birds used social information to learn the materials were safe to interact with, even though they did not use social information to immediately solve the task.

Additional comments

Specific comments - numerous, though mostly minor and easily corrected.

1. Line 18: At this point it is unclear what the authors mean by ‘socially obtained information’, a broad term, which would argue against there being only one study conducted with a less social corvid. See also lines 56-57 and 65-67.
2. Line 20: All instances of i.e. should occur in parentheses. For example, “use social information (i.e., information made available by others).”
3. Line 46-47: Change to “These findings suggest that species differences in ecology determine whether particular behaviours are expressed, even though most or all species likely share the underlying cognitive capability required for the behaviour”
4. Lines 52-55: Could the authors expand on this argument? Why would a behaviour or cognitive ability be lost simply from lack of a selection pressure, rather than a selection pressure against the trait (i.e., neural cost for producing the ability)?
5. Line 57: “(species that live in groups of at least pairs year-round)” This definition of a ‘social’ species would include most corvid species considered less social, such as Clark’s nutcrackers and, arguably, Eurasian jays. Instead of making a strict cut-off point for social/asocial species, I recommend describing sociality as a spectrum within corvids, such that Eurasian jays are relatively less social compared to other more social corvid species. I recommend the authors use ‘relatively asocial’ or ‘less social’ throughout the manuscript.
6. Line 71: Given the lack of corvid studies, I recommend the authors include non-corvid research that could speak to whether living in large groups matters for social learning (e.g., Lefebvre, Palameta, & Hatch, 1996, Behaviour).
7. Lines 76-77: The authors should add a bit more information here about the social life of Eurasian jays. In particular, whether the birds form life-long or just seasonal monogamous pairbonds. From this sentence, it is unclear whether the mated pair separates after the breeding season.
8. Lines 84-85: This sentence is unclear, what specific ‘roles’ can the jays play while caching and pilfering? How does dominance rank affect their behaviour?
9. Line 92: “…primarily solitary species.” This seems to describe their social life as more extreme than in other papers (e.g., Shaw & Clayton, 2012), which indicate these birds are often found in pairs, as well as small, loose flocks. Is it fair to describe this species as primarily solitary?
10. Line 142: Were the birds housed socially or individually within the outdoor enclosure?
11. Line 145: Change to “(one pair, three groups of 3 birds, and one group of 4 birds)”, or some alternative to make this more readable.
12. Line 146: The wording “within inside test compartments” is a bit awkward. Do the authors mean these compartments were indoors?
13. Line 155: Could the authors comment on why such young birds were tested? How does this compare to the other species studied? What is the developmental trajectory of Eurasian jays, particularly in regards to neophobia?
14. Line 155: I know this group studies (mental) time travel, but I suspect “November 2016” is a typo.
15. Lines 159-161: This description could be more precise. I recommend changing to “The testing apparatus was a clear Perspex ‘object insertion’ apparatus consisting of a tube and a collapsible platform. Objects could
be dropped into the tube, causing the collapsible platform on the bottom of the tube to release from a small magnet holding it in place. Once released from the magnet, a food reward was dispensed to the subject”.
16. Lines 161-163: It is unclear from this sentence what all these parts are (plastic rings, removable platform) and how they relate to what is seen in Figure 1 and 2. This could be easily addressed by labelling these parts in the figures.
17. Line 165: What materials were the objects made from?
18. Lines 168-172: From this paragraph it is unclear how the ‘trained’ group differs from the ‘control’ group, which made the subsequent procedures difficult to follow. From this section, it seems both groups try to learn the task without any assistance. The reader only learns the distinction between these two groups a few paragraphs later.
19. Line 175: Be more specific here in terms of what stage of the apparatus the birds were given. I recommend adding this level of detail throughout the methods (e.g., lines 224, 235).
20. Line 176-177: Was the apparatus absent? It would also be helpful if the authors provided brief statements explaining why each procedure was conducted.
21. Lines 179-181: Add a statement here explaining the purpose of the removable platform.
22. Line 184: What is a session? How many trials are in each session? How many sessions were given during a day? Without an explanation of this term on first use, the rest of the methods were difficult to follow. It is not until experiment 2 (lines 252-253) that sessions are explained, so it is unclear whether this description is also true for experiment 1.
23. Line 185: Why was it necessary to train the birds to this level of difficulty in the task rather than simply using stage 1? If the observers only ever get to attempt the task with a stage 3 apparatus, why were demonstrations of previous stages given? Seems like these stage 1 and stage 2 demonstrations would only interfere with the learning needed for stage 3, as the demonstrated behaviours can’t be used as a solution for stage 3.
24. Line 187: Change to “No birds spontaneously solved the task, therefore they were randomly assigned…”
25. Line 194: “trained the ‘trained group’” is a bit awkward. Could reword to “Birds in the ‘trained group’ were shaped to solve the task…”
26. Line 201-206: What criteria determined when birds moved between stages?
27. Lines 207-210: How was Homer’s performance as a demonstrator? Was he 100% accurate or were the observers exposed to failed demonstrations as well? This information should be provided for experiment 2 as well.
28. Lines 213-214: I am unclear on the order of events here. Did the birds watch stage one 40 times, then stage two 40 times, etc.? Or did the birds watch stage 1 once, then stage 2 once, then stage 3 once, and then that sequence was repeated 40 times?
29. Lines 217-219: What was the spacing between birds like? Would observers be wary of attending to the demonstrator for fear of potential aggression? Did the observers have sufficient visual access to see all aspects of the apparatus? Importantly, could they see that the object drop was always associated with the release of the food reward?
30. Lines 218-219: The birds had free visual access between compartments, but were they enclosed within a compartment or free to come and go? Also, it is unclear whether other observers were present in other compartments or even within the same compartment during demonstrations. If observers were not isolated from one another during the demonstrations, then how did the authors ensure the birds attended sufficiently to the demonstrations, rather than to their distracting social environment? Descriptions on lines 96-97 and 387 imply this was measured, but I can’t find a direct measure of attention reported.
31. Line 223: How did these 40 demonstrated ‘solves’ map onto the 16 demonstration sessions? Over how many days were the observers tested? Did they experience a demonstration prior to every test?
32. Lines 241-242: The wording here is ambiguous and makes it sound like the two cups were attached to each other – was this the case? Or were they both attached to the wooden board? Or each attached to its own metal rod, so that each cup could be moved up and down the rod independently?
33. Line 255: Was Homer given 4 more sessions in addition to the 3 it took to reach criterion, or did he receive 4 total sessions?
34. Line 260: Were the birds used in the second task from the group used in the first task?
35. Lines 321-323: This section should be removed. The video could be supplied as supplementary material and referenced somewhere in the text.
36. Lines 330-331: “No observer or control birds ever came close to solving the task.” This sentence is colloquial and could be removed.
37. Line 335: “Individuals in the observer group were not more likely to interact with the apparatus or object”. More appropriate wording here would be ‘Individuals in the observer group did not make more touches to the apparatus or object…’ as this stays closer to the actual measured variables.
38. Lines 339-341: From Figure 4 this looks to be a case of statistical but not experimental significance. The interaction looks to be the result of trial 3 in which the two groups diverge. I recommend the authors mention the results are likely due to a seemingly random blip during trial 3 (see also lines 389-392).
39. Line 343: This supplementary table does not seem necessary. I think a simple statement in the text that the best model was selected using AIC should suffice.
40. Lines 346-348: “Birds in the trained group required more insertions to solve the task”. Insertions are how the birds solve the task, so do the authors mean the trained group took more ‘trials’ to solve the task? See also lines 306-307, 395, and Figure 5 (caption and y-axis label). Or if I am misunderstanding this point, revise the methods and analysis sections so that this point is clarified.
41. Lines 352-353: “2 of 7 jays…” The ‘2’ should be written out if starting a sentence.
42. Lines 443-441: This is unclear – what does copying a mate choice mean? Who is copying from whom? Is there evidence to suggest this takes place rather than using the publically available physical indicators of mate quality? I recommend replacing this section with an expanded discussion of the individual characteristics of the demonstrator used (i.e., Homer) that could have influenced the jays’ decisions (e.g., dominance, personality, information producer/scrounger). This would keep the discussion closer to the experiment conducted and to points that have more literature to support them.
43. Line 446: Should read ‘a sibling’ not “as sibling”.
44. Lines 446-448: Should mention the statistical limitations of your study here. It was not designed to be able to address questions of relatedness, and there were not enough birds to conduct a conclusive analysis.
45. Line 450: The authors should provide a brief description of each of the stages listed here.
46. Lines 452-453: “…information on <25% and <5% of occasions…” It is unclear why two values are listed here.
47. Lines 454-456: Include the caveat here that social species also don’t show much evidence for observational learning in this task.
48. Lines 457-458: I would change this to “Eurasian jays did not use social information to immediately copy the choice of the demonstrator…” as you can’t conclude they didn’t use social information, they just didn’t use it in the behaviours you measured on a single trial (see also lines 365-366). Also consider revising the manuscript title in a similar way.
49. Table 1: I recommend removing this table. Information from this table could be put solely in the text, or in the figure caption of figure 2.
50. Table 3 and 4: These tables are either labeled wrong or ordered incorrectly.
51. Table ‘4’ (counterbalancing table): This table is unnecessary and could be removed. This information is easily conveyed in the text with a statement that side was counterbalanced.
52. Table ‘3’ (glm results): The table is unnecessary as this analysis could easily (and more appropriately) be described in the text of the results.
53. Figure 1 and 2: I recommend merging these two figures. Either include the dimensions on panel (a) of Figure 2, or add Figure 1 as a panel in Figure 2.

---

## Round 0.2 · Minor Revisions

First, apologies for taking longer than usual to make a decision on this MS. I’ve now had a chance to carefully read the revised manuscript and your response to the reviewers. Thank you for carefully acknowledging each of the reviewers’ many thoughtful comments. I think this paper will make a very nice contribution to the literature. I just have a few extremely minor grammatical issues I’d like you to attend to before I can formally accept the MS.

Please check that you’ve always included the latin name of the species at first mention (e.g., line 110).
Write out numbers <10 in full.
Write out “minutes” in full on line 350.
On line 433, could “solves” be changed to “solutions with”. It sounds awkward as is.
On line 596, the “to” at the end of the sentence should be “the”
The animal ethics statement now appears to refer only to Study 2, but presumably it applies to both experiments? Can you place that as a general statement earlier in the MS, or revise to say “these experiments were…”
Also be consistent in referring to Experiments, rather than “Studies” given that there was a manipulation here.
Delete the “only” on line 897.
Lines 902-903; Not a complete sentence. Perhaps change “Though” to “Although, we reiterate…object-dropping task used in Exp.1”.. without the “used” it sounds like you tested social species in this current exp.
Move the “only” from line 908 to between “one study” on line 909.

---

## Round 0.3 · accepted · Accept

Thank you for attending quickly to these last very minor editorial issues. I am now happy to accept your paper for publication and think it will make a nice complement to other work in this area.